# Sequence-Specific Features of Short Double-Strand, Blunt-End RNAs Have RIG-I- and Type 1 Interferon-Dependent or -Independent Anti-Viral Effects

**DOI:** 10.3390/v14071407

**Published:** 2022-06-28

**Authors:** Abhilash Kannan, Maarit Suomalainen, Romain Volle, Michael Bauer, Marco Amsler, Hung V. Trinh, Stefano Vavassori, Jana Pachlopnik Schmid, Guilherme Vilhena, Alberto Marín-González, Ruben Perez, Andrea Franceschini, Christian von Mering, Silvio Hemmi, Urs F. Greber

**Affiliations:** 1Department of Molecular Life Sciences, University of Zürich, 8057 Zürich, Switzerland; abhilashkannan2987@gmail.com (A.K.); maarit.suomalainen@mls.uzh.ch (M.S.); romain.volle@uzh.ch (R.V.); mbauer@rockefeller.edu (M.B.); marco.amsler@hotmail.com (M.A.); htrinh@genezen.com (H.V.T.); atariw@gmail.com (A.F.); christian.von.mering@mls.uzh.ch (C.v.M.); silvio.hemmi@mls.uzh.ch (S.H.); 2Neurimmune AG, Wagistrasse 18, 8952 Schlieren, Switzerland; 3Laboratory of Virology and Infectious Disease, The Rockefeller University, New York, NY 10065, USA; 4Genezen, 9900 Westpoint Dr, Suite 128, Indianapolis, IN 46256, USA; 5Division of Immunology, University Children’s Hospital Zürich, 8032 Zürich, Switzerland; stefano.vavassori@kispi.uzh.ch (S.V.); jana.pachlopnik@kispi.uzh.ch (J.P.S.); 6Faculty of Medicine, University of Zürich, 8006 Zürich, Switzerland; 7Departamento de Física Teórica de la Materia Condensada, Universidad Autónoma de Madrid, E-28049 Madrid, Spain; guilhermevilhena@gmail.com (G.V.); ruben.p.perez@gmail.com (R.P.); 8Condensed Matter Physics Center (IFIMAC), Universidad Autónoma de Madrid, E-28049 Madrid, Spain; 9Department of Macromolecular Structures, Centro Nacional de Biotecnología, Consejo Superior de Investigaciones Científicas, Cantoblanco, E-28049 Madrid, Spain; amaring1@jhmi.edu; 10Center for Genomic Science of IIT@SEMM, Fondazione Istituto Italiano di Tecnologia, 20139 Milano, Italy; 11Swiss Institute of Bioinformatics, 1015 Lausanne, Switzerland

**Keywords:** short double-strand blunt-end RNA, RIG-I, interferon, DNA virus, RNA virus, RNA therapy, antiviral agents, adenovirus, influenza virus, SARS-CoV-2

## Abstract

Pathogen-associated molecular patterns, including cytoplasmic DNA and double-strand (ds)RNA trigger the induction of interferon (IFN) and antiviral states protecting cells and organisms from pathogens. Here we discovered that the transfection of human airway cell lines or non-transformed fibroblasts with 24mer dsRNA mimicking the cellular micro-RNA (miR)29b-1* gives strong anti-viral effects against human adenovirus type 5 (AdV-C5), influenza A virus X31 (H3N2), and SARS-CoV-2. These anti-viral effects required blunt-end complementary RNA strands and were not elicited by corresponding single-strand RNAs. dsRNA miR-29b-1* but not randomized miR-29b-1* mimics induced IFN-stimulated gene expression, and downregulated cell adhesion and cell cycle genes, as indicated by transcriptomics and IFN-I responsive Mx1-promoter activity assays. The inhibition of AdV-C5 infection with miR-29b-1* mimic depended on the IFN-alpha receptor 2 (IFNAR2) and the RNA-helicase retinoic acid-inducible gene I (RIG-I) but not cytoplasmic RNA sensors MDA5 and ZNFX1 or MyD88/TRIF adaptors. The antiviral effects of miR29b-1* were independent of a central AUAU-motif inducing dsRNA bending, as mimics with disrupted AUAU-motif were anti-viral in normal but not RIG-I knock-out (KO) or IFNAR2-KO cells. The screening of a library of scrambled short dsRNA sequences identified also anti-viral mimics functioning independently of RIG-I and IFNAR2, thus exemplifying the diverse anti-viral mechanisms of short blunt-end dsRNAs.

## 1. Introduction

Viruses infecting humans and animals are a significant health risk, especially those that lack effective treatment and are resistant against medications or vaccines but also emerging pathogens, for which there is no treatment and little knowledge. Broad antiviral in vitro strategies have explored short synthetic double-stranded RNAs (dsRNAs) mimicking small interfering RNAs (siRNAs) or microRNAs (miRs) against virus infections. siRNAs are 21–23 nucleotide long dsRNAs, typically with two nucleotide overhangs at their 3′-ends. The siRNA guide strand interacts stably with RNA-induced silencing complex (RISC) and the complex targets mRNAs with complementary sequences to degradation [1], thus achieving gene knockdown or near knock-out (KO) phenotypes in the transfected cells. Genome-wide screens with synthetic siRNAs have identified numerous pro- and anti-viral host factors for a number of different viruses, for recent examples, see for example, [2,3].

Similar to siRNAs, miRs inhibit gene expression by inducing mRNA degradation and/or translational repression via RISC but, unlike siRNAs, miR effects require only a limited complementarity to the target and involve only the 5′ two to eight nucleotides on the guide strand, the so-called seed sequence [1]. Cell endogenous miRs are transcribed as precursors and post-transcriptionally processed to yield about 22–24 nucleotide long mature duplex miRs with typically two-nucleotide long 3′ overhangs [1]. Their nomenclature follows a sequential numbering scheme in the order of their discovery in a particular organism, such as *Homo sapiens* (hsa) [4]. If the seed sequence is identical, but the precursors are distinct and arise from different genes, the number is extended by a letter, such as hsa-miR-29a, hsa-miR-29b, or hsa-miR-29c [5]. An additional number distinguishes miRs from different genomic loci encoding identical mature miR sequences, such as hsa-miR-29b-1 or hsa-miR-29b-2. Furthermore, the miR from a strand opposite to that encoding an abundantly expressed miR precursor is annotated with an asterisk, such as hsa-miR-29b-1*, and strand identifiers are used in the case of miR expression levels are unknown, that is, 5p and 3p for a miR precursor are derived from the 5′-strand and the 3′-strand, respectively. 

Pioneering as well as large scale siRNA screens have been conducted with several different viruses, for example the negative-stranded segmented RNA virus influenza A virus [6,7,8,9,10,11,12,13], the positive-sense RNA viruses severe acute respiratory syndrome coronavirus (SARS-CoV) and SARS-CoV-2 [14,15,16], and the double-stranded DNA virus adenovirus (AdV) [2,10,17]. However, there has generally been a limited overlap in the pro- or anti-viral genes identified for a given virus in the different screens [18,19,20]. Typically, viruses take advantage of cellular miRs or encode viral miRs [21]. For example, IAV upregulates miR expression profiles, including hsa-miR-101, hsa-miR-193b, hsa-miR-23b, and hsa-miR-30e* [22], or is boosted upon forced expression of miR340-5p, which attenuates cellular antiviral immunity [23]. Adenoviruses evolved mechanisms to counteract the cellular RNAi machinery by producing a set of short dsRNAs that can function as miRs and are derived from the noncoding virus-associated RNAs [24,25,26,27,28]. SARS-CoV-2 expresses small viral noncoding miR-like transcripts, which have the potential to interfere with host transcripts involved in IFN signaling [29,30], and COVID-19 leads to altered miR expression with prognostic potential [31]. Emerging evidence, however, also indicates that a range of cellular hsa-miRs, including miR-219a-2-3p, miR-30c-5p, miR-378d, miR-29a-3p, miR-15b-5p, and miR-15b, intercept SARS-CoV-2 gene expression by targeting viral transcripts [32]. 

Given these intriguing attributes of miR, genome-wide miR mimic screens or profiling of miRs in virus-infected cells have the potential to identify miRs that enhance or suppress virus life cycle by direct targeting of virus RNAs or by modifying host gene expression, as exemplified by hepatitis C virus [33]. However, the interpretation of transfection results with synthetic dsRNA mimics is not always straightforward, despite the fact that siRNA or miR mimics have proved to be valuable tools in virus research. For example, since siRNAs and miRs both use RISC, a majority of hits from siRNA screens can be from off-targets due to siRNAs acting as miRs, rather than solely silencing the mRNA of interest exhibiting perfect complementarity [34,35,36]. Furthermore, dsRNAs in cells are under innate immune surveillance. Synthetic siRNAs have been described to induce interferon (IFN) response via pattern recognition receptors (PRRs), including Toll-like receptor 7 (TLR7), TLR8, and retinoic acid-inducible gene 1 (RIG-I), notably independent of the RISC-mediated target knockdown activity [37,38,39,40]. TLR7 and TLR8 detect single-stranded (ss) RNAs located in endosomal compartments, where the majority of transfected synthetic siRNAs and miRs accumulate, at least when lipid-based transfection reagents are used [41,42]. The denaturation of duplex siRNA/miR mimics in the endosomes enhances activation of TLR7/TLR8, as concluded from observations that one of the ssRNA strands commonly yields as good or even better activation of TLR7/TLR8 than the equivalent duplex RNA [37,38,39,43]. Ligand-engaged TLR7/TLR8 then activate IFN production via myeloid differentiation primary response 88 (MyD88) signaling pathways, and secreted IFN in turn induce expression of IFN stimulated genes (ISGs), which are capable of restricting replication of a broad range of viruses [44,45]. 

In contrast to TLR7/TLR8, RIG-I is a dsRNA sensor located in the cytoplasm and, akin to MDA5, uses mitochondrial antiviral-signaling protein (MAVS) to activate IFN production [45,46]. RIG-I and MDA5 have nonredundant functions, where RIG-I is activated by dsRNAs containing a blunt-end and 5′-triphosphate group (5′ppp), such as panhandle structures of viral genomes [47] or copy-back defective interfering particles [48,49], and thereby contribute to infection variability [50], while MDA5 requires several hundred base pairs of duplex RNA stem structures, which occur as replication intermediates, for example in picornaviruses or coronaviruses [51,52]. Unlike RIG-I and MDA5, LGP2 (laboratory of genetics and physiology 2) does not directly activate MAVS but rather tunes the sensor functions of RIG-I and MDA5 [48,53]. Here, we present a synthetic 24-bp blunt-end dsRNA with a 5′-OH structure based on the hsa-miR-29b-1* (hsa-miR-29b-1-5p), which directly or indirectly activates RIG-I signaling and induces broad anti-viral effects to cultured and primary transfected cells in a sequence-dependent manner. The antiviral effects of miR29b-1* mimic dsRNA go along with IFN induction and are abrogated in RIG-I or IFNAR2 KO cells. These results contrast RIG-I and IFNAR2 independent antiviral effects of scrambled dsRNA mimics, indicating sequence specific antiviral functions of short blunt-end dsRNAs. 

## 2. Materials and Methods

### 2.1. Cells

Human lung epithelial carcinoma A549, HeLa, and HEK293T cells were from the American Type Culture Collection and maintained in growth medium consisting of DMEM (Sigma-Aldrich, St. Louis, MO, USA, D6429) supplemented with 7.5% fetal calf serum (FCS; Gibco/Thermo Fisher Scientific, Waltham, MA, USA, 10270106) and 1% non-essential amino acids (Sigma-Aldrich, M7145). Non-transformed human dermal fibroblast lines 120.03 and 105.03 were obtained from donors at the Division of Immunology at the University Children’s Hospital Zürich (Zürich, Switzerland; ClinicalTrials.gov. NCT02735824; IRB reference: PB_2016_02280) and have been described previously [54]. Cells were maintained in DMEM (ThermoFisher Scientific, 31966-021) supplemented with 10% FCS, 1% non-essential amino acids, 1% GlutaMAX (Thermo Fisher Scientific, 35050-061), 1 mM sodium pyruvate (Sigma-Aldrich, S8636), and 100 µg/mL kanamycin sulfate (Thermo Fisher Scientific 15160-054). The clone 120.03 is from a healthy donor and 105.03 has a mutation in CD48, but this gene is not expressed in fibroblasts [55]. A549 cells expressing the SARS-CoV-2 receptor angiotensin-converting enzyme 2 (A549-ACE2 cells) were used as described [56]. A549 RIG-I and A549 IFNAR2 knockout (KO) cells were generated from ATCC A549 cells by CRISPR/Cas9 technology using a lentivirus vector for a doxycycline-inducible expression of Cas9 and a U6-promoter-driven expression for the gRNA. Lenti-tet-eCas9 was generated by modifying the previously described Lenti-eCas9 [2]. Briefly, the EF-1α core promoter in front of eCas9 was replaced with a tight tetracycline response element (TRE) promoter. Downstream of eCas9, an EF-1α core promoter was inserted in front of a tetracycline-controlled transactivator (rtTA) in frame with the P2A sequence and the puromycin resistance cassette. Two different RIG-I KO clones were produced using gRNA templates 5′-GGATTATATCCGGAAGACCC-3′ (sg1) and 5′-TGACTGCCTCGGTTGGTGTT-3′ (sg2) targeting exons one and eight, respectively. The sequence used as a gRNA template for IFNAR2 KO cells was 5′-CCGTCCTAGAAGGATTCAGC-3′ targeting exon five. The gRNAs were designed using the online tool GuideScan (http://www.guidescan.com accessed on 1 September 2020) [57]. The sequence was cloned into the BsmB1 site of the lenti-tet-eCas9 and lentivirus vectors were produced according to the instructions of the Zhang lab [58]. A549 cells were transduced with lentivirus vectors at low multiplicity of infection and two days after transduction cells were switched to a medium containing 2 µg/mL puromycin and 1 µg/mL doxycyclin. After five days the surviving cells were expanded using the normal growth medium without puromycin and doxycyclin. For subcloning the transduced cells were passed through a 0.4 µm cell strainer (Falcon, 352340) following trypsinization and seeded at a low density on 15 cm tissue culture plates. Well isolated cell colonies were subsequently picked using sterile cloning cylinders (Merck, CLS31666). After expansion, RIG-I clones were initially tested for editing by analyzing RIG-I induction after the treatment of cells with recombinant interferon alpha 2 (IFNα2; Biogen) using Western blots and the rabbit anti-RIG-I antibody D14G6 (cell signaling; data not shown), whereas IFNAR2 clones were tested for Mx1 induction after IFNα2 treatment using Western blots and mouse anti-Mx1 antibodies (kindly provided by Jovan Pavlovic, University of Zürich, Switzerland; data not shown). The genomic DNA from promising clones was isolated using Qiagen Dneasy Blood and tissue kit (69506), and the region containing the sgRNA target site was amplified by PCR using Q5 DNA polymerase (New England BioLabs, M0491) and oligo pairs 5′-AGCACCCTTAAGCAAGTCAC-3′ and 5′-GCGGAGGGAAACGAAACTAG-3′ (RIGI KO sg1), 5′-GTTAGAGAGGCTAAAAAATG-3′ and 5′-GCTTCAGAAGAGCTATAATC-3′ (RIGI KO sg2), or 5′-CTCCTGGAGCAGACATTATG-3′ and 5′-AAACCGAAAACAGCAGTTCC-3′ (IFNAR2 KO) as primers. The PCR fragments were gel-purified using Promega Wizard SV gel and PCR clean-up kit (A9282) and sequenced by Sanger sequencing. The sequences were analyzed using the TIDE web tool (http://shinyapps.datacurators.nl/tide; version 3.2.0 accessed on 16 September 2020 for IFNAR2 KO, version 3.3.0 accessed on 23 October 2021 for RIG-I KO sg1 and on 26 November 2021 for RIG-I KO sg2) [59]. The TIDE analyses indicated that RIG-I KO sg1 clone used in the study was composed of a 7 base pair (bp) deletion (25.8%), a 4 bp deletion (28.6%), and a 1 bp insertion (27.7%), whereas the RIG-I KO sg2 clone had 1 bp insertion (75.9%) and a 31 bp deletion (19.5%). The PCR for IFNAR2 KO clone yielded two fragments, the expected 718 bp fragment and a shorter about 400 bp fragment. The TIDE analyses indicated that the long fragment carried deletions of 7 bp (91%) and 3 bp (1.8%), whereas the shorter fragment consisted of a 329-bp deletion. 

### 2.2. Viruses

AdV-C5 virus was grown in A549 cells and purified on CsCl gradients as previously described [60]. The influenza A virus (IAV-H3N2 X31 strain) used in the experiments was kindly provided by Yohei Yamauchi (University of Bristol, Bristol, UK). SARS-CoV-2 “Wuhan” (TAR clone 3.3, München-1.1/2020/929) was obtained from Dr. Volker Thiel (University of Bern, Bern, Switzerland) [61]. SARS-CoV-2 experiments were conducted at BSL3 conditions (Institute of Pathology, University of Zürich, Zürich, Switzerland), as described [56]. 

### 2.3. dsRNA Oligos

The dsRNA miR mimics for the human hsa-miR-29b-1-5p (miR29b-1*; MIMAT0004514) were obtained from Qiagen (Qiagen AG, Hombrechtikon, Switzerland, miScript miR mimic MSY0004514; the miScript product line has been discontinued), Dharmacon (Dharmacon, Horizon Discovery Ltd., Cambridge, UK, miRIDIAN mimic C-301150-01-0002), or custom-made by Microsynth (Microsynth AG, Balgach, Switzerland). The Qiagen miScript mimics are unmodified blunt-end dsRNAs with 5′-OH and 3′-OH, whereas Dharmacon miRIDIAN mimics contain 3′-overhangs, a monophosphate in the 5′-antisense, and carry proprietary ON-TARGET modifications on the passenger strand complementary to the antisense strand. The custom-made Microsynth mimics were designed either as unmodified blunt-end dsRNAs with 5′-OH and 3′-OH or as dsRNAs with 3′-UU overhangs. The Rand dsRNA with a randomized miR29b-1* sequence was a custom-made miScript mimic from Qiagen or a blunt-end unmodified mimic from Microsynth. The Scr dsRNAs were custom-made Qiagen miScript miR mimics. The blunt-end dsRNAs for testing the importance of dsRNA bending by AU-tract sequence [62] (miR29b-1*_sAU2 and miR29b-1*_sAU3) were from Microsynth. The sequences for the dsRNAs used in the study can be found in Appendix A. 

### 2.4. Transfections and Infections

Infection assays were carried out in 96-well imaging plates (Greiner Bio-One GmbH, Frickenhausen, Germany, 655090). For mimic transfections, the following amounts of reagents were used per well: dsRNAs (to yield final concentration of 10 nM) and 0.15 µL Lipofectamine RNAimax (Thermo Fisher Scientific, 13778030) were mixed in 20 µL Optimem (Thermo Fisher Scientific, 11058-021), incubated at room temperature (RT) for 20 min and 10’000 A549 cells or non-transformed fibroblast cells were added in 80 µL growth medium. After 24–28 h post transfection, virus was added to cells in growth medium containing 1% penicillin-streptomycin (Sigma, P0781). The input virus amounts were adjusted so that the infection index (number of infected cells) was on linear range with respect to input virus amounts, i.e., two-fold dilutions of input virus yielded about two-fold lower number of infected cells. A549 AdV-C5- infections were fixed with 3% PFA in phosphate-buffered saline (PBS) for 30 min at room temperature at 24–26 h post infection (pi), whereas non-transformed fibroblast infections were fixed at 46 h pi with virus inoculum medium being replaced by fresh penicillin-streptomycin containing growth medium at 22 h post virus addition. Infection efficiencies were scored by immunostaining using anti-VI antibody [63] and secondary Alexa Fluor 488-conjugated goat anti-rabbit antibodies (Thermo Fisher Scientific A-11034, 2 µg/mL) as described in [64]. Nuclei were stained with 4′,6-diamidino-2-phenylindol (DAPI, 1 µg/mL solution in PBS; Sigma D9542). IAV infections were fixed at 12 h pi and analyzed by immunostaining using a mouse anti-NP HB65 antibody (kindly provided by Yohei Yamauchi, University of Bristol, UK) and secondary Alexa Fluor 488-conjugated goat anti-mouse antibodies (Thermo Fisher Scientific A-11029, 2 µg/mL). SARS-CoV-2 infections were carried out in A549-ACE2 cells and infections were analyzed at 24.5 h pi using an anti-nucleocapsid antibody (Rockland 200-401-A50) and secondary Alexa Fluor 488-conjugated goat anti-rabbit antibodies.

For the analysis of virus progeny production, dsRNA transfections were carried out as described above, except a 24-well dish was used with final dsRNA concentration of 10 nM, 0.6 µL Lipofectamine RNAimax and 40,000 A549 cells per well. Infection was started at 24 h post transfection using~1.5 infectious units of AdV-C5 per cell. Virus inoculum was removed 7 h pi, cells were washed twice with excess volumes of DMEM to remove residual unbound virus and incubation was continued in growth medium containing penicillin/streptomycin for additional 17 h, 41 h or 65 h before collecting medium- and cell-associated virus progeny. Cell debris was removed from the medium sample by centrifugation at 215× *g* for 5 min. For the collection of cell-associated viruses, cells were first lifted up by a brief incubation in PBS-0.5 mM EDTA and cells were recovered by centrifugation at 215× *g* for 5 min. The cell pellet was resuspended in 10 mM TRIS-HCl pH 8.1 containing 1 mM phenylmethylsulfonyl fluoride, the suspension was freeze-thawed three times using liquid nitrogen and 37 °C water bath and extracted with Tris-saturated Freon 113 (Sigma-Aldrich, product line discontinued). The aqueous phase containing the virus particles was collected following centrifugation at 1900× *g* at +4 °C for 5 min. The medium- and cell-associated infectious units were titrated on HeLa cells scoring number of infected cells at 24 h pi using anti-VI staining.

### 2.5. Imaging and Image Analysis

Infected 96-well plates were imaged with a Molecular Devices automated ImageXpress Micro XL wide-field imaging system using 10 × Plan Fluor (numerical aperture 0.3) and a single focal plane for all channels corresponding to a middle section of nuclei. Images were analyzed either by a custom-programmed MatLab (The Mathworks, Inc., Bern, Switzerland) routine (provided by Martin Engelke, the Greber lab) or by a CellProfiler pipeline (CellProfiler version 2.2.0, http://cellprofiler.org) [65]. Nuclei were segmented using the DAPI image and the mean AdV-C5 protein VI and influenza nucleoprotein intensities over the DAPI mask were determined (both AdV-C5 and influenza nucleoprotein accumulate in the nucleus) as described [66]. For SARS-CoV-2, mean cytoplasmic intensities for the nucleocapsid signal were determined by CellProfiler using a cytoplasmic mask obtained by extending the DAPI mask by ten pixels while the nuclear area was excluded from this cytoplasmic mask. Infection efficiencies were scored as infection index, i.e., the number of infected cells over the total number of cells analyzed. Non-infected control wells were used to determine the threshold for an infected cell, threshold being either the maximum value from non-infected cells or, if the maximum values were clearly outlier values (e.g., due to autofluorescent dust particles in the wells), then the 99.5% cut-off value from the non-infected cell population was used. The MatLab routine calculated mean infection indexes and standard deviations from technical replicates, whereas Knime Analytics Platform (https://www.knime.com/knime-analytics-platform, version 3.3.2, Knime AG, Zurich, Switzerland) was used to calculate infection indexes from the CellProfiler data. GraphPad Prism 7 (GraphPad Software, La Jolla, CA, USA) was used to create the graphs. Representative images were processed in Fiji [67], applying the same changes in brightness and contrast to all images in the series.

### 2.6. Gene Expression Profiling of miR29b-1*-Transfected A549 Cells

Qiagen miR29b-1* miR mimic (final concentration 10 nM) was mixed with 3.3 µL of Lipofectamine RNAimax in 442 µL Optimem and the mix was incubated at room temperature for 45 min before addition of A549 cells (~560,000 cells in 1.75 mL growth medium) and the mix was plated on 6-well tissue culture plates. Non-targeting siRNA from Qiagen (AllStars, 1027281) was used as a control. Three technical replicates were used for the miR29b-1* and four technical replicates for the control. Total RNA was purified at 72 h post transfection using Ambion mirVana miR Isolation Kit (Thermo Fisher Scientific, AM1560). The quality of RNA was assessed by measuring the RNA integrity number (RIN) using Bioanalyzer 2100 (Agilent Technologies, Santa Clara, CA, USA) and was confirmed to be above seven in all the samples. Total RNA purity was determined using Nanodrop ND-1000 (Thermo Scientific, Waltham, MA, USA), where the A260:A280 ratio was >2.0 and A260:A230 > 2. The labeling of the RNA samples for gene expression analysis was performed using the protocol for the One-Color Microarray-Based Gene Expression analysis together with the Low Input Quick Amp Labeling kit (Agilent Technologies). For the labeling, 1.5 μL of diluted RNA samples (150 ng total RNA) were used and one-color labeling was carried out with Cy3 (Cyanine 3 CTP, Agilent). Subsequent steps of the gene expression profiling, including hybridization, washing, and measurement was carried out at the Functional Genomics Center (FGCZ), Zürich University, Switzerland, according to the manufacturer’s protocol. The chips used were custom-made 4x44K arrays (Chip type: 020887, Agilent) designed in-house and contained probes for all human genes. The whole-genome expression data were analyzed by use of the Bioconductor package in R, version 3.0.1. The LIMMA package was used to identify differentially expressed genes in miR29b-1*-transfected cells. Obtained p-values were adjusted for multiple testing using the false discovery rate (FDR) method; −0.5 > Log_2_FC > 0.5 and FDR < 0.05. The pathway enrichment analyses of differentially expressed genes were carried out using MetaCore (Thomson Reuters/Clarivate, Cortellis, Philadelphia, PA, USA) [68]. Appendix A lists the top 50 significantly up- and down-regulated host genes in miR29b-1*-transfected cells in comparison to the control transfection. Appendix A lists the significant-scoring pathway maps of up- and down-regulated genes in miR29b-1* transfected cells in comparison to the control transfection.

### 2.7. Measurement of IFN Production from dsRNA-Transfected Cells

Qiagen miR29b-1* mimics (final concentration 2.5 nM) or Microsynth miR29b-1* mimics (final concentration 10 nM) were transfected into A549 cells using Lipofectamine RNAimax and 96-well plate format as described above. Cleared culture supernatants from the cells were collected at 24 h, 48 h, and 60 h post transfection or at 47–48 h, as indicated, and titrated on HEK293T reporter cells. These cells were seeded the day before (200,000 cells/well in a 24-well plate) and co-transfected with a reporter plasmid (200 ng) encoding Firefly (FF) luciferase (pGL3-Mx1-FFLuc) under the IFN-inducible Mx1 promoter, a plasmid (50 ng) encoding Renilla (REN) luciferase under constitutive SV40 promoter (pRLSV40-RenillaLuc; both luciferase plasmids were kindly provided by Jovan Pavlovic, University of Zürich, Switzerland [69]), and a carrier plasmid (750 ng) using 2 µL/well jetPEI (Polyplus 101-10N) or 1 µL/well jetPRIME (Polyplus 101000027) according to a protocol recommended by the manufacturer. The transfection medium was removed at 6–6.5 h post transfection and replaced by a standard growth medium (final volume 500 µL) containing 100 µL of the clarified culture supernatants from the dsRNA-transfected cells or different concentrations of recombinant IFNα2 (1.25 U, 2.5 U, 5 U, and 10 U). Cells were lysed at 20–24 h post IFNα2/clarified culture supernatant stimulation and FF- and REN-luciferase activities were measured using Tecan Infinite M200 plate reader and Promega dual-luciferase reporter assay kit (E1910) according to the manufacturer’s instructions.

### 2.8. siRNA Transfections

Two rounds of reverse transfection were used. The first round was siRNA transfection (final concentration 20 nM) on a 24-well plate using 0.9 µL Lipofectamine RNAimax and 40,000 or 80,000 (siZNFX1) A549 cells/well, as described above. All other siRNAs were siGenome SMARTpool siRNAs from Dharmacon, except siZNFX1, which were Dharmacon ON-TARGET plus SMARTpool siRNAs. The transfection medium was replaced by a fresh standard growth medium containing penicillin/streptomycin at 22 h post transfection and a second round transfection was carried out 26 h later. For the second round, cells were trypsinized and reverse transfected with siRNAs (20 nM final concentration) plus Rand or miR29b-1* dsRNAs (final concentration 10 nM; Microsynth dsRNAs for siZNFX1 and Qiagen mimics for the others), 0.15 µL Lipofectamine RNAimax, and ~10,000 cells/well and were seeded on a 96-well imaging plate. At 24–25 h post second transfection, cells were infected with AdV-C5 and analyzed at 24–25 h pi by immunostaining for the late protein VI as described above. Knockdown levels were controlled by RT-qPCR as described below. For ZNFX1, cell lysates were prepared 49 h after the first round of transfections and intracellular levels of ZNFX1 were analyzed by Western blotting using rabbit anti-ZNFX1 (Abcam ab179452, EPR12330, Abcam plc, Cambridge, UK) and a secondary HRP-conjugated anti-rabbit antibody (Cell Signaling Technology, Danvers, MA, USA, #7074). The antibody for the gel loading control was mouse anti-alpha/beta tubulin (Amersham N357) and the secondary was a HRP-conjugated anti-mouse antibody (Cell Signaling Technology, #7076).

### 2.9. RT-qPCR

The total RNA was extracted from siRNA transfected A549 cells (40,000 cells/well) using the miRVANA kit as per manufacturer’s protocol (Thermo Fisher Scientific, AM1560). The purity and concentration of total RNA was determined using Nanodrop ND-1000 (Thermo Scientific, Waltham, MA, USA), that is, A260:A280 ratio > 2.0; A260:A230 > 2. 300 ng of total RNA was reverse transcribed using 50 nM specific RT primers in 20μL reaction mixtures also containing 1× First-strand buffer (SuperScript^TM^ III Reverse transcriptase kit, Invitrogen Cat. No. 18080044), 0.25 nM deoxynucleoside triphosphate (dNTP) mix, 10 mM DTT, 4 U/μL RNAseOUT (Invitrogen, Cat. No. 10777019) RNase inhibitor, and 25 U/μL SuperScript III Reverse Transcriptase (Invitrogen, Cat. No. 18080044). The primers used were designed to specifically reverse transcribe the genes of interest and are listed in Appendix A. Reaction mixtures were incubated at 16 °C for 30 min, 30 °C for 30 s, 42 °C for 30 s, and 50 °C for 1 s. These steps were repeated in 60 cycles and the samples finally incubated at 85 °C for 5 min. The cDNA was amplified by PCR using the Power SYBR Green Master mix (Applied Biosystems). PCR reactions were carried out with an ABI Prism 7900HT Sequence Detection System (Applied Biosystems) using the following cycle conditions: 95 °C for 5 min, followed by 40 cycles at 95 °C for 15 s and at 60 °C for 1 min. Relative miRNA levels (2−ΔCt) were determined by comparing the PCR quantification cycle (Ct, determined with the Software SDS 2.2.1). The threshold cycle number for the PCR product (mRNA levels) were normalized to that of *EEF1A1*, *TBP*, *TFRC*, *ACTB*, and/or *GAPDH* mRNAs and expressed as mean relative mRNA levels (n = 3 per treatment).

## 3. Results

### 3.1. A Short Blunt-End dsRNA Oligo with hsa-miR-29b-1* Sequence Reduces AdV-C5 Infection

The starting point for the current study was a miR profiling of AdV-C5- and AdV-B3-infected A549 cells with custom-made Agilent microarrays (results to be published elsewhere). We were primarily interested in miRs that were downregulated in both AdV-C5 and AdV-B3 infections, since downregulated miRs represent potential anti-viral miRs [70]. Our attention was drawn to hsa-miR-29b-1-5p (miR29b-1*), which was among the top downregulated miRs in our study, as well as in a previously published miR profiling data of AdV-C5-infected A549 cells [28]. However, endogenous levels of miR29b-1* in A549 cells are low (see our results and [28]), and thus it is unclear whether the observed downregulation of this miR in AdV infections is of biological relevance. 

To explore possible relevance, we used synthetic double-stranded miR29b-1* mimics from three different suppliers. The mimic from Qiagen was a chemically unmodified, blunt-end dsRNA with 5′-OH and 3′-OH. The one from Dharmacon had 3′ overhangs and its passenger strand was chemically modified to reduce RISC incorporation and the 5′-antisense contained a monophosphate. The mimics were transfected to A549 cells at the concentration of 10 nM using Lipofectamine RNAimax, cells were infected with AdV-C5 for 24 h, and scored by staining for the viral late protein VI. A Qiagen mimic with a randomized miR29b-1* sequence (Rand) was used as a control (Appendix A). The Qiagen miR29b-1* mimic, but not Rand, reduced the number of VI-positive cells in comparison to mock transfection, that is, RNAimax without mimics (Figure 1A). In contrast, Dharmacon miR29b-1* mimic had essentially no effect on infection efficiency (Figure 1B). Furthermore, cultures transfected with the Qiagen miR29b-1* mimic had reduced overall cell numbers in comparison to control transfections, a phenotype that was not evident in the Dharmacon miR29b-1* transfection (Figure 1A,B). 

To understand the conflicting results between the Qiagen and Dharmacon mimics, we used synthetic blunt-end or 3′ overhang but chemically unmodified mimics with 5′-OH from a third manufacturer (Microsynth) and tested them in A549 cells. The results indicated that the blunt-end Microsynth miR29b-1* mimic, but not the mimic with overhangs, was a good inhibitor of AdV-C5 infection (Figure 1B). The blunt-end miR29b-1* mimic potently inhibited AdV-C5 infection in two non-transformed human fibroblast cell clones as well (Figure 1C). Strong reduction in progeny production was observed in miR29b-1*-transfected A549 cells in comparison to RNAimax-treated or Rand-transfected control cells (Figure 1D). Taken together, these results suggest that the ds blunt-end miR29b-1* mimics might not act akin to endogenous miRs but reduce AdV-C5 infection efficiency by another mechanism. 

### 3.2. Blunt-End dsRNA miR29b-1* Mimic Induces Interferon Response When Transfected into A549 Cells

We next tested the effect of the blunt-end miR29b-1* mimic on two other viruses, the human influenza A (IAV) virus (X31) and SARS-CoV-2. The IAV infection of A549 cells was scored at 12 h p.i. by immunofluorescence using antibodies against the virus nucleoprotein (NP). A clear reduction in NP-positive cells was observed in miR29b-1*-transfected cells, but not in the Rand transfection (Figure 2). Similarly, when SARS-CoV-2 infection was analyzed in A549 cells expressing the viral receptor angiotensin-converting enzyme (ACE)-2, anti-nucleocapsid staining at 24.5 h pi indicated a clear reduction in infection efficiency upon miR29b-1*, but not Rand transfections. 

To understand the basis of the broad anti-viral effect of the miR29b-1* blunt-end mimic, we profiled cellular gene expression changes using whole genome microarrays from Agilent Technologies. Cells transfected with the Qiagen non-targeting siRNA (siAllstar) were used as a control. Since, according to the manufacturer, siAllstar produces minimal transcriptome changes, siAllstar was chosen as a control. The cells transfected with 10 nM miR29b-1* and Lipofectamine RNAimax contained 7617 statistically significant changed transcripts (FDR values < 0.05), compared to control transfection representing about 38% of the total interrogated host transcripts in the microarray. Accordingly, Appendix A lists the top 50 up- and down-regulated host genes. Taking advantage of the MetaCore system, we conducted bioinformatics pathway enrichment analysis with the differentially expressed genes based on their log2 fold change values and p-values [68]. Most of the upregulated genes were linked to innate immune response and IFN–induced genes (Figure 3A and Appendix A). The gene ontology (GO) of the downregulated genes revealed strong links to cell adhesion and the cell cycle. Overall, the results suggest that the anti-viral effects of the blunt-end miR29b-1* mimic could be due to induction of IFN response.

To probe whether a blunt-end miR29b-1* dsRNA mimic induced IFN secretion, we analyzed the cell culture supernatants at 24, 48, and 60 h post transfection on indicator 293T cells, which expressed Firefly luciferase under the IFN-inducible Mx1 promoter and Renilla luciferase under the constitutive SV40 promoter. Culture supernatants from the miR29b-1*-transfected cells activated the IFN-dependent Mx1-promoter, but essentially no activation was evident with the supernatants from non-transfected or Rand-transfected cells (Figure 3B). To confirm these results, we produced IFNAR2 knockout (KO) A549 cells by CRISPR/Cas9-mediated gene editing. KO was verified by sequencing the genomic target region of the sgRNA used. In the IFNAR2-KO cells AdV-C5 infection was no longer attenuated by miR29b-1* mimic and indistinguishable from Rand transfection (Figure 3C) Likewise, reduction of cell numbers was less pronounced, suggesting that the effect on cell viability/growth was due to IFN response, rather than general toxicity.

### 3.3. Anti-Viral Effects of Blunt-End miR29b-1* Mimic Are Mediated by RIG-I

The Qiagen miR29b-1* mimic used in this study is a commercial dsRNA molecule with unknown quality control for contaminating ssRNA structures. The Microsynth dsRNA miR29b-1* blunt-end mimics were made by simple hybridization of the “sense” and “antisense” strands. To ascertain that the anti-viral effect of the blunt-end miR29b-1* mimic was due to the dsRNA structure of the mimic and not to contaminating ssRNAs, we transfected the “sense” and “antisense” ssRNA oligos of miR29b-1* individually to A549 and tested their effect on AdV-C5 infection. As shown in Figure 4A, AdV-C5 infection in ssRNA-transfected cells was comparable to that of Rand transfection. Furthermore, siRNA-mediated knockdown of MyD88, a downstream effector of the ssRNA sensors TLR7/TLR8 [45], did not reduce anti-viral effects of the blunt-end double-stranded miR29b-1* mimic (Appendix A).

The cytoplasmic dsRNA sensor RIG-I has been linked to IFN response induced by short blunt-end dsRNAs, e.g., [40,71,72]. When intracellular levels of RIG-I or its downstream effector MAVS were reduced by siRNAs in A549 cells, transfection of miR29b-1* no longer affected AdV-C5 infection compared to Rand transfected cells (Figure 4B). In contrast, the miR29b-1* mimic still efficiently reduced AdV-C5 infection when the dsRNA sensor MDA5 or TRIF, the downstream effector of the dsRNA sensor TLR3 [73], were knocked down by siRNAs (Appendix A). To confirm the importance of RIG-I, we produced two CRISPR/Cas9-edited RIG-I KO A549 cell clones using different guide RNAs (sgRNA1 and sgRNA2). Knockouts were verified by sequencing the genomic target region of the sgRNAs. The AdV-C5 infection phenotype was comparable in miR29b-1*- and Rand-transfected sgRNA1-KO cells, and similar results were obtained with the sgRNA2 RIG-I KO clone (Figure 4C). Furthermore, as shown for sgRNA1 KO cells in Figure 4D, miR29b-1* mimic did not induce IFN secretion in cells devoid of RIG-I. Thus, the anti-viral effects of blunt-end dsRNA miR29b-1* mimic are, directly or indirectly, mediated by RIG-I. 

RIG-I (*DDX58*) is an IFN-induced gene but expressed at very low levels in uninduced cells, for example, as in [74]. ZNFX1 (zinc finger NFX1-type containing 1) is a constitutively expressed dsRNA sensor in A549 cells, and it has been postulated to be an early responder for dsRNAs, initiating IFN production and thus enhancing RIG-I intracellular levels and RIG-I-mediated anti-viral responses [54,74]. We used siRNAs directed against ZNFX1 to test whether this sensor plays a role in the anti-viral phenotype of the blunt-end miR29b-1* mimic. As shown in Appendix A, miR29b-1* mimic transfection significantly reduced AdV-C5 infection in siRNA-treated cells, despite efficient ZNFX1 knockdown. The results suggest that ZNFX1 is likely not involved in the antiviral effects induced by miR29b-1* in A549 cells. 

### 3.4. Identification of miR-29b-1* Mimic-Related and Unrelated dsRNAs Inhibiting AdV-C5 Infection

The different phenotypes of Rand and miR29b-1*-transfections indicate that blunt-end short dsRNAs can differ in their anti-viral effects, suggesting that the anti-viral effects of miR29b-1* are sequence-dependent. miR29b-1* mimic contains a centrally positioned AUAU-motif, which constitutes a 4 bp-long AU-tract, i.e., dsRNA sequences of alternating adenines and uracils. Molecular dynamics simulations and atomic force microscopy (AFM) have indicated that dsRNA AU-tracts induce a bend in the molecule due to a local compression of the major groove [62,75]. Such bend has been detected also in miR29b-1* [62]. In order to test whether the AU-tract-induced bend is important for anti-viral effects of the blunt-end miR29b-1* mimic, we reshuffled the central miR29b-1* sequence to disrupt the AUAU-motif. Accordingly, the modified sequences of the two mimics, miR29b-1*_sAU2 and miR29b-1*_sAU3, are listed in Appendix A and also shown in Figure 5. miR29b-1*_sAU2 and miR29b-1*_sAU3 still efficiently reduced AdV-C5 infection in wild type parental A549 cells, but not in RIG-I KO A549 cells (Figure 5A,B). IFN secretion was slightly lower from miR29b-1*_sAU2- or miR29b-1*_sAU3-transfected wild type A549 cells than from miR29b-1*-transfected cells (Figure 5C). Furthermore, similar to miR29b-1*, AdV-C5 infection was fully recovered in miR29b-1*_sAU2-transfected A549-IFNAR2 KO cells, and significant recovery was observed in the miR29b-1*_sAU3 transfection (Figure 5D). These results suggest that the central AU-tract is not critically important for RIG-I- and IFN-mediated anti-viral effects of the blunt-end miR29b-1* mimic. 

We next designed additional 33 blunt-end, 24-nucleotide long dsRNAs with scrambled sequences (scr1-scr33; Appendix A) to explore the frequency of anti-viral effects of dsRNA transfections. To reduce possible miR-like effects of the RNAs, all of them had the same “seed sequence” (i.e., nucleotides 2–8) as the sense-strand of Rand, namely UUAUGCG (Figure 6). Twenty-two of the dsRNAs (scr 1–14, 20–27) retained the base composition of miR29b-1*, whereas the other eleven did not (scr 15–19, 28–33). The anti-viral phenotype of all these dsRNAs was tested in A549 cells with Lipofectamine RNAimax-mediated transfection. As shown in Figure 6A, several of the Scr dsRNAs reduced AdV-C5 infection at least to some degree, four of them as efficiently as miR29b-1*, namely Scr 20, 24, 32, 33. Remarkably, Scr 20, 24, 32, 33 dsRNAs retained their anti-viral effects in both RIG-I KO and IFNAR2 KO A549 cells, unlike the miR29b-1* (Figure 6B,C). Notably, Scr 20 and 24 have the same base composition as miR29b-1*, namely 41% GC content, and the other two have an increased GC content >55%, suggesting that short blunt-end dsRNAs habor a range yet-to-be characterized anti-viral features. Together, the results show that antiviral effects of short blunt-end dsRNAs can be rather common, depending on particular features of the dsRNA, and occur directly or indirectly through RIG-I and IFN or are independent of RIG-I and IFNAR2. 

## 4. Discussion

RNA therapeutics have come of age, as mRNA vaccination is highly effective against COVID-19, and therapies with RNA encoding viral and bacterial surface proteins or effectors in cancer therapeutics are increasingly considered in clinical applications, for a review, see [76]. Challenges, however, remain, including immunogenicity, accurate and effective RNA delivery in noninflammatory settings, or maintenance of sufficient expression levels over prolonged periods of time [48,77]. Arguably, RNA immunogenicity has been best studied with viruses, which produce defective and truncated dsRNAs, and activate sensor proteins, including TLRs and RIG-I-like receptors, such as RIG-I and MDA5 [48]. Accordingly, dsRNAs have been found to induce a plethora of effects in transfected cells as well as organisms, including IFN stimulation by poly I:C ribonucleotides, RNA interference with siRNAs, broad transcriptional effects with miRs, or interference by aptamers, short RNA molecules with a defined secondary structure binding to selected targets with high affinity [78,79,80]. MicroRNAs and siRNAs have a wide potential for applications although their effects on gene expression in general are rather modest [81,82]. 

Initial fortuitous observations with select siRNAs or miR mimics alerted to the possibility of triggering an IFN response. This was followed by a search for features in siRNAs/miR mimics that mediate activation of PRR, for a review, see [83]. Early descriptions of TLR7/TLR8 stimulation by synthetic siRNA pointed to specific sequence motifs in siRNAs, for example 5′-GUCCUUCAA-3′ and 5′-UGUGU-3′ [38,39]. Yet, a subsequent broad analysis indicated that TLR7/TLR8 stimulation by standard 21mer dsRNAs with two base 3′-overhangs and unmodified nucleotides is not restricted to just a few specific sequence motifs but is positively correlated with high U numbers in the U-rich strand of the duplex and negatively with the hybridization strength of the duplex strands [37,84]. The latter most likely reflects the importance of endosomal duplex RNA denaturation for effective IFN signaling. Accordingly, replacing U residues with N1-methylpseudouridine has been shown to reduce innate immune sensing of synthetic mRNAs transfected into cells [85,86]. The enrichment of uracil in miR29b-1* (10 U out of 24 nucleotides), however, does not explain the antiviral effects of our miR29b-1*, since the Rand miR, for example, had the same number of U residues and was not capable of protecting cells from SARS-CoV-2, IAV, or AdV infections. 

Interestingly, miR-29 family members are thought to have tumor suppressor functions and affect gene expression in multiple cancers through various processes, including epigenetics, proteostasis, metabolism, proliferation, apoptosis, metastasis, fibrosis, angiogenesis, and immunomodulation [87]. Our transfected short blunt-end dsRNAs miR-29b-1* mimic, however, antagonized virus infections independent of canonical miR functions. The miR29b-1* mimic, as well as its derivatives miR29b-1*_sAU2 and _sAU3, exerted their effects against AdV-C5 in a manner depending on RIG-I and IFN production at concentrations of 10 nM or below, conditions that preclude a generic IFN response [88,89]. Under these conditions, the effects of the mimics on IFN stimulation and the anti-viral effects were dependent on a particular sequence signature, but apparently independent of GC content. 

It currently remains unknown whether endogenous levels of the miR29b-1* would suffice to induce an antiviral response against AdV-C5 comparable to the transfected miRs. Earlier RNAseq analyses of AdV-C5 infected A549 cells uncovered only low levels of miR-29b-1* [28]. In addition, immuno-isolated RISC contained an overrepresentation of only those mRNAs, which were complementary to the seed sequence of viral miRs of VA-RNA I and II, but not those complementary to cellular miRs, including miR29b-1*. This suggests that endogenous miR-29b-1* likely has no major role for mRNA silencing in AdV-C5 infected cells, unlike VA I and II miRs. This is in agreement with our observation that transfection of ssRNA-sense or -antisense strand of the mature miR29b-1* had no effect on AdV-C5 infection or cell number, unlike the dsRNA miR29b-1* mimic. We infer that the downregulation of cell adhesion and cell cycle transcripts by miR29b-1* is not the result of miR29b-1* incorporation into RISC but rather a signature of the antiviral state induced by IFN, reviewed in [90,91], and shown for the cell adhesion gene COL21A1 in human osteoblast differentiation [92]. Thus, IFN induction is likely responsible for the reduced cell numbers on the plates upon miR29b-1* treatment, as compared to the Rand mimics. However, we note that the miR29b-1* mimic had a potent anti-viral effect in non-transformed fibroblasts and also in A549 cells against IAV without leading to significant reduction in cell numbers. Thus, cell toxicity is unlikely to be the main reason for anti-viral effects of the miR29b-1* mimic. 

Invariably, the antiviral effects of our miR29b-1* mimics required blunt-ends and depended on RIG-I and IFNAR2. RIG-I is known to be strongly activated by short, blunt-end stem-loop RNAs with ≥ten base pairs of perfect complementarity and harboring a 5′-triphosphorylated terminus, as found in panhandle structures of negative-strand viruses, or by short 19-25 base-paired dsRNAs with a blunt-end 5′-tri- or diphosphorylated terminus [71,72,93,94,95]. Short blunt-end dsRNAs with a 5′-monophosphate or a 5′-OH structure are less effective compared to dsRNAs with a 5′-tri- or diphosphorylated structure [95,96,97], but 5′-monophosphate is sufficient for at least partial RIG-I activation [39,40,94,98]. Short blunt-end dsRNAs bind to RIG-I in vitro regardless of whether they contain 5′-OH or 5′-monophosphates. The 5′-phosphate on the dsRNA promotes innate immune activation and IFN stimulation in transfected murine L929 cells or mouse embryo fibroblasts (MEF) but not in MEF lacking RIG-I [98]. We postulate that our dsRNA mimics become 5′-monophosphorylated upon entering cells, and hence are a strong enough PAMP to activate RIG-I. A possible kinase for 5′-OH phosphorylation of miR29b-1* is Clp1, which phosphorylates short blunt-end dsRNAs in vitro [99]. 

Our 24-nucleotide long miR29b-1* dsRNA is a strong activator of RIG-I and IFN, strictly depends on blunt-ends and is not active with 3′-overhangs or modified passenger strands. This is in line with dsRNAs characterized by others [40,94,96,97]. Accordingly, the antiviral effects of the dsRNA miR29b-1* mimic were sequence-dependent, although the essential features of the sequence motifs necessary to activate RIG-I are still unclear. Interestingly, molecular dynamics simulations indicate that AU-tract motives in the context of GC-rich sequences give rise to dsRNA bending, owing to the collapse of the major groove located away from the sugar backbone, likely due to the lack of a hydrogen bond in AU compared to GC base pairs [62,75]. Additional simulations showed that changes in the major groove widths are correlated to minor groove changes by distinct geometrical alterations in the base pairs [100]. This implies that changes in the major groove width could affect the antiviral innate immune signalling from dsRNA binding proteins, including RIG-I, which recognizes the phosphate backbone in the minor grove in a sequence independent manner, reviewed in [48]. However, the ablation of the AUAU-motif in miR29b-1* did not abrogate the antiviral effects of miR29b-1*, indicating that motifs other than AUAU also mediate antiviral effects. This is not surprising in light of the notion that only 7 out of 33 scrambled dsRNAs had strictly no effect on AdV-C5 infection and cell numbers (scr 2, 3, 8, 17, 19, 22, 31), while 22 had intermediate inhibitory effects (scr 1, 4-7, 9-16, 18, 21, 23, 25-30), and 4 had antiviral effects as strong as miR29b-1* but independent of RIG-I and IFNAR2, namely scr 20, 24, 32, 33. Notably, two of the strong antiviral dsRNAs had the same base composition as miR29b-1*, and two had increased GC contents, suggesting that the strand melting properties of the dsRNAs are not solely decisive for the antiviral action. Besides being a direct activator of RIG-I, it is possible that the miR29b-1* mimic acts as a decoy and indirectly activates RIG-I by competing with endogenous RIG-I activators for binding to proteins, which normally shield dsRNA and thereby limit RIG-I activation. Examples for endogenous dsRNAs activating RIG-I are the signal recognition particle RNA RN7SL1, a 5′-triphosphorylated RNA polymerase III transcript, or circular RNAs [101,102,103]. It is now possible to apply systematic permutations of the antiviral dsRNAs and conduct gene expression profiling or genome-wide KO studies to identify further sensors and effectors besides RIG-I and IFN. This will enhance insights into host sensing mechanisms and the nature of antiviral PAMPs as well as host mechanisms antagonizing the sensing of PAMPs. Such studies may then take the field beyond the simple question of “self” versus “nonself” recognition in context of dsRNAs. 

## Figures and Tables

**Figure 1 viruses-14-01407-f001:**
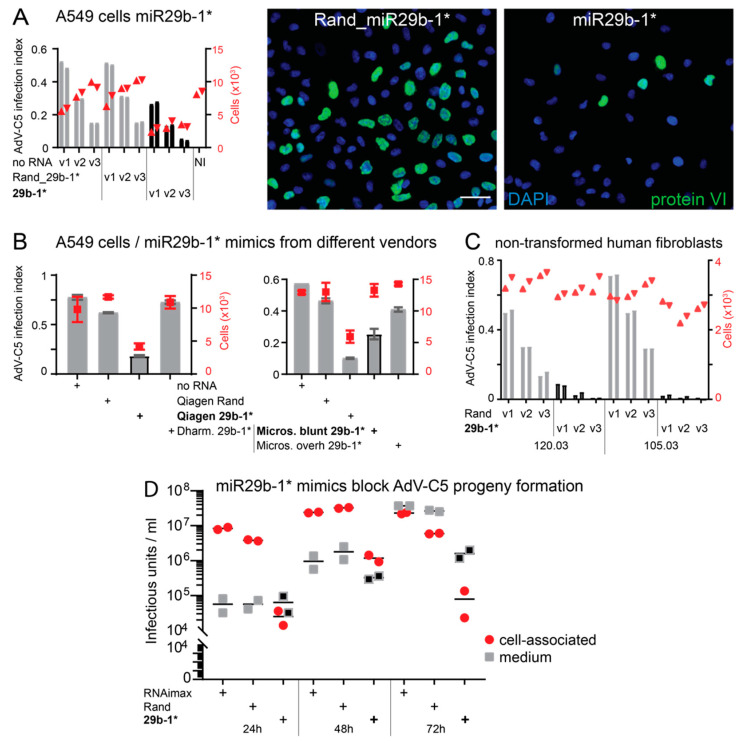
A short blunt-end dsRNA with miR29b-1* sequence reduces AdV-C5 infection efficiency. (**A**). Qiagen miR29b-1* mimic (29b-1*), but not a control mimic with a randomized miR29b-1* sequence (Rand), reduces AdV-C5 infection efficiency in A549 cells. RNAimax was used for the mimic transfections. No RNA refers to non-transfected cells and v1, v2, and v3 to three different two-fold dilutions of input virus. Infection efficiencies were scored by immunostaining for the late virus protein VI, and infection index (left *y*-axis, bar plot) refers to the fraction of VI-positive cells over total number of cells analyzed (right *y*-axis, scatter plot red triangles). The two technical replicates are shown separately. Representative images from the v2-input virus infection are shown on the right-hand side for cells transfected with the Rand and miR29b-1* mimics. Scale bar = 50 µm. (**B**). A549 cells transfected with miR29b-1* mimics from different vendors display different infection phenotypes. The blunt-end dsRNA miR29b-1* mimics from Qiagen (Qiagen 29b-1*) and Microsynth (Micros. blunt 29b-1*) reduced AdV-C5 infection efficiency, but Dharmacon (Dharm. 29b-1*) or Microsynth (Micros. overh 29b-1*) dsRNA mimics with 3′ overhangs did not significantly deviate from the results obtained with the Rand. Shown are mean values from three technical replicates (infection index bar plot, red squares cell numbers). The error bars represent standard deviation. The left and right panels are from two independent experiments. (**C**). The blunt-end miR29b-1* mimic potently inhibits AdV-C5 infection in non-transformed human fibroblasts. The two technical replicates are shown separately. 120.03 and 105.03 represent cells from two different donors. Grey bars represent infection index, red symbols cell numbers. (**D**). The blunt-end miR29b-1* mimic significantly reduces virus progeny production in A549 cells. At the shown time points, cell-associated and extracellular progeny virions were collected and titrated on HeLa cells by staining for the late protein VI. RNAimax refers to mock transfection with transfection reagent but without RNA mimics. The titer of medium-associated progeny from miR29b-1* mimic-transfected cells is highlighted as black squares.

**Figure 2 viruses-14-01407-f002:**
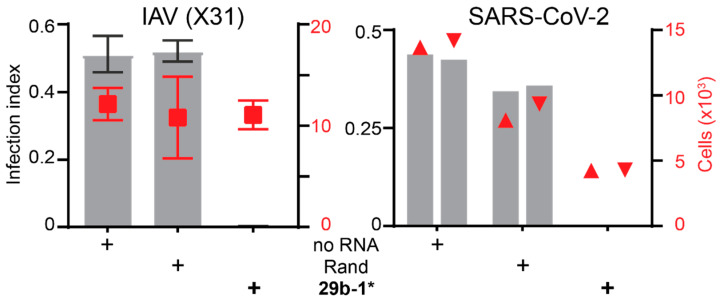
The blunt-end dsRNA miR29b-1* mimic has broad anti-viral effects in A549 cells and efficiently reduces IAV and SARS-CoV-2 infections. The IAV infection efficiency was scored by immunostaining for the viral nucleoprotein. The SARS-CoV-2 experiment was carried out in A549-ACE2 cells and infection efficiency was scored by immunostaining against the viral nucleocapsid protein. The bars represent infection index (left *y*-axis) and the cell numbers analyzed (red squares) are shown as a scatter plot (right *y*-axis). Shown are for IAV mean values from three technical replicates and the error bars represent standard deviation. The two technical replicates in the SARS-CoV-2 experiment are shown separately. No RNA refers to non-transfected cells, Rand to transfection with the randomized miR29b-1* sequence mimic, and 29b-1* to transfection with the miR29b-1* mimic. Rand and miR29b-1* mimics were from Qiagen in the IAV experiment and from Microsynth in the SARS-CoV-2 experiment.

**Figure 3 viruses-14-01407-f003:**
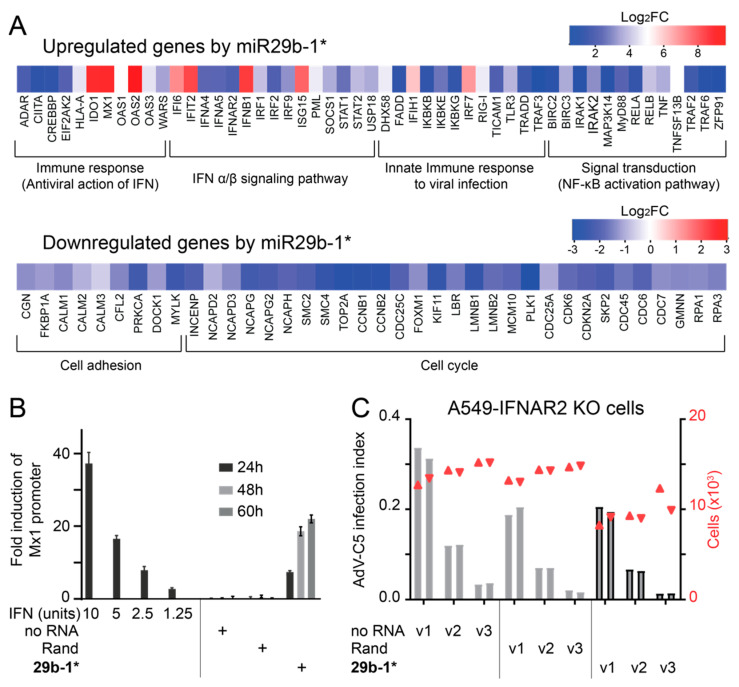
The blunt-end dsRNA miR29b-1* mimic induces IFN response in cells and this response mediates the anti-viral effects of the oligo in A549 cells. (**A**) Agilent whole genome microarray analysis of gene expression changes in Qiagen miR29b-1*-transfected A549 cells in comparison to control cells, which were transfected with Qiagen non-targeting dsRNA siAllstar (10 nM). Shown is a pathway enrichment analysis of significantly (*p*-value ≤ 0.05) up-regulated and down-regulated functional pathways, the heatmap illustrating log_2_ values of gene expression changes for the indicated genes within the pathways. (**B**) A549 cells transfected with blunt-end miR29b-1* mimic (29b-1*, 2.5 nM), but not those transfected with randomized miR29b-1* sequence mimic (Rand), secrete type I interferons. Clarified culture supernatants were collected from the cells at 24 h, 48 h, and 60 h post transfection and titrated on 293T indicator cells expressing Firefly luciferase under the type I IFN-inducible Mx1 promoter (jetPEI transfections). The *y*-axis shows normalized Firefly luciferase activities in cell extracts expressed as fold-changes in comparison to values obtained from untreated indicator cells. Indicated amounts of recombinant IFNα2 were used as controls. The bars represent mean values from three technical replicates with standard deviation. No RNA indicates culture supernatant from non-transfected A549 cells. Rand and miR29b-1* mimics were obtained from Qiagen. (**C**). The anti-viral effect of the blunt-end dsRNA miR29b-1* mimic is ablated in A549 IFNAR2-KO cells. miR29b-1*-transfected A549-IFNAR2 knockout cells were infected with AdV-C5 and infection efficiencies were scored by immunostaining for the late virus protein VI. Infection index (left *y*-axis, bar plot) refers to the fraction of VI-positive cells over total number of cells analyzed (right *y*-axis, scatter plot) and v1, v2, and v3 to three different two-fold dilutions of input virus. The two technical replicates are shown separately. Grey bars represent infection index, red symbols cell numbers.

**Figure 4 viruses-14-01407-f004:**
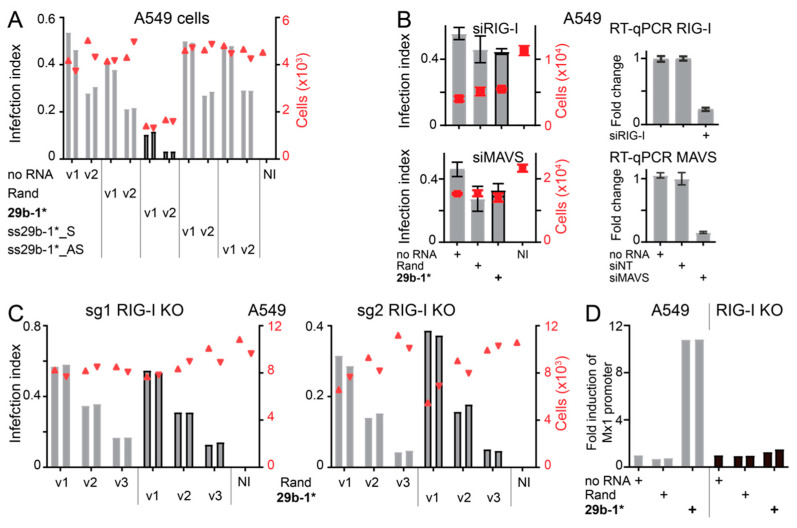
RIG-I mediates anti-viral effects of the blunt-end dsRNA miR29b-1* mimic. (**A**) Anti-viral effects of the blunt-end miR29b-1* mimic (29b-1*) require dsRNA structure since the individual ssRNA oligos of the mimic (ss29b-1*_S and ss29b-1*_AS) do not reduce AdV-C5 infection efficiency. A549 cells were infected with AdV-C5 and infection efficiencies were scored by immunostaining for the late virus protein VI. The left *y*-axis (bar plot) shows the infection index, the right *y*-axis (scatter plot, red triangles) shows the number of cells analyzed, and v1 and v2 refer to two different two-fold dilutions of the input virus. The two technical replicates are shown separately. No RNA indicates cells treated with the RNAimax transfection reagent alone and Rand refers to cells transfected with the mimic with randomized miR29b-1* sequence. NI is the non-infected control. (**B**) siRNA-mediated knockdown of RIG-I or MAVS equalized AdV-C5 infection in miR29b-1* and Rand mimic-transfected A549 cells. Infection efficiencies were scored as in (**A**) and the values shown represent the mean from three technical replicates with standard deviation (grey bars infection index, red squares cell numbers). siRNA-mediated knockdown efficiencies were scored by RT-qPCR. siNT refers to a control, non-targeting siRNA. Rand and miR29b-1* mimics were obtained from Qiagen. (**C**) The anti-viral effect of the blunt-end dsRNA miR29b-1* mimic (10 nM final concentration) is ablated in A549-RIG-I knockout cells. Results from two different knockout clones created by two different guide RNAs (sg1 and sg2) are shown. The AdV-C5 infection efficiency was scored as in (**A**). v1, v2, and v3 represent three different input virus amounts with two-fold dilutions. (**D**) No significant IFN secretion from miR29b-1*-transfected A549-RIG-I KO (sg1) cells. Clarified culture supernatants were collected from the transfected cells at 48 h post-transfection and scored for IFN on 293T indicator cells as described in legend to Figure 3B. The values were normalized to that of mock transfection (RNAimax only) and the two technical replicates in the assay are shown separately.

**Figure 5 viruses-14-01407-f005:**
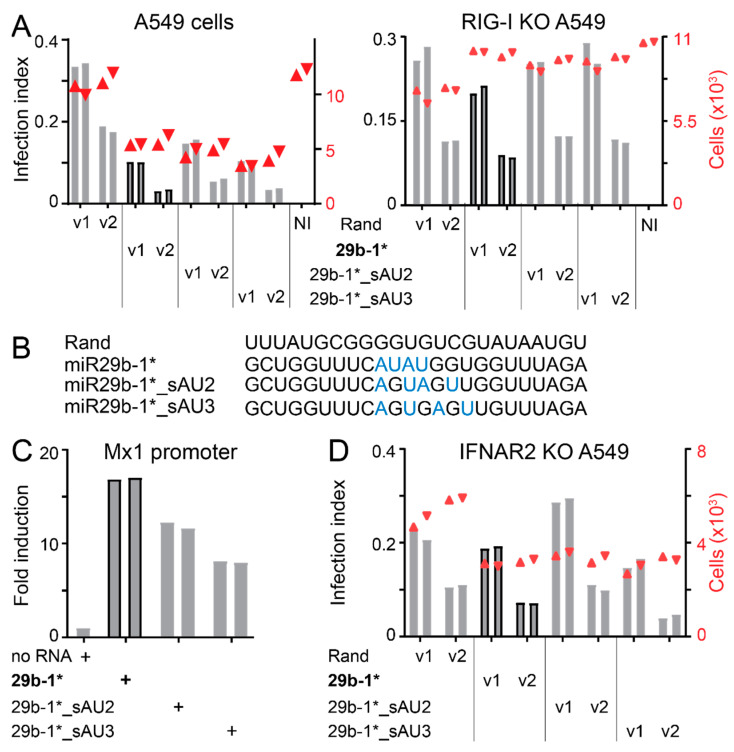
Altering the centrally positioned AUAU-tract in miR29b-1* does not affect the anti-viral potency of the mimics. (**A**,**B**) The blunt-end dsRNA miR29b-1* (29b-1*) contains a centrally positioned AUAU-tract that induces a bend into the RNA molecule. Modified blunt-end dsRNA miR29b-1* mimics with a disrupted central AUAU-tract (29b-1*_sAU2 and 29b-1*_sAU3) still retain anti-viral effects against AdV-C5 in parental but not in RIG-I knockout A549 cells. Infection efficiencies were scored by immunostaining for the late virus protein VI. Infection index (left *y*-axis, bar plot) refers to the fraction of VI-positive cells over total number of cells analyzed (right *y*-axis, scatter plot, red triangles). The two technical replicates are shown separately, and v1 and v2 refer to two different two-fold dilutions of the input virus. Rand refers to cells transfected with the mimic with randomized miR29b-1* sequence and NI is the non-infected control. Sequences of the mimics are shown in the 5′ to 3′ orientation leaving the reverse complement strand out. (**C**) Central AUAU-tract motif is not important for IFN induction. The experiment was carried out as described in legend to Figure 3B. dsRNA were used at 10 nM final concentration. No RNA refers to medium from cells treated with the transfection reagent RNAimax alone. (**D**) Significant infection recovery in miR29b-1*_sAU2- or miR29b-1*_sAU3-transfected IFNAR2 KO A549 cells. Infection efficiencies were scored as described in panel (**A**).

**Figure 6 viruses-14-01407-f006:**
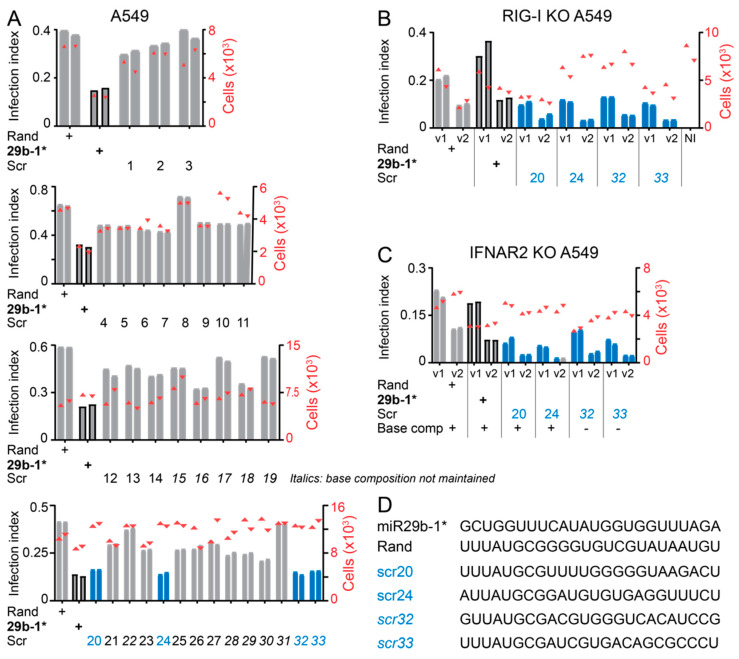
Short blunt-end dsRNAs with randomized sequences commonly show anti-viral effects but not always via RIG-I and IFN. (**A**) AdV-C5 infection efficiency in wild type A549 cells transfected with scrambled, blunt-end 24-nucleotide long dsRNAs. The sequences of the mimics used are listed in Appendix A. Infection efficiencies were scored by immunostaining for the late virus protein VI and infection index (left *y*-axis, bar plot) refers to the fraction of VI-positive cells over total number of cells analyzed (right *y*-axis, scatter plot red triangles). The two technical replicates are shown separately. (**B**,**C**) Test of the potent anti-viral scrambled dsRNAs in A549 RIG-I KO (**B**) and IFNAR2 KO (**C**) cells. Analyses were carried out as in (**A**), except that two different two-fold dilutions of the input virus were used. The experiments shown in Figure 5D and Figure 6C are from the same 96-well plate and therefore the Rand and 29b-1* controls are the same in these two figures. Grey bars represent infection index, red symbols cell numbers. (**D**) List of sequences of the guide strands (5′ to 3′-orientation) of dsRNAs miR29b-1*, Rand, and scr 20, 24, *32, 33,* where numbers are in italics indicate dsRNAs with a base composition distinct from miR29b-1*, Rand, scr 20, and 24.

## Data Availability

The data and scripts used to create the figures in this manuscript (raw microscopy images, plate overviews, CellProfiler pipelines, MatLab scripts, and result Excel files) are deposited at Zenodo.org (https://doi.org/10.5281/zenodo.6641856). Due to the large size of the data of Figure 1D and Figure 6A, only plate overviews are uploaded to Zenodo. The original raw microscopy images are available from the authors upon request.

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
