# Peer review of "Sequence-Specific Features of Short Double-Strand, Blunt-End RNAs Have RIG-I- and Type 1 Interferon-Dependent or -Independent Anti-Viral Effects"

_viruses, 2022, doi:10.3390/v14071407_

Round 1

Reviewer 1 Report

In their m/s, Kannan etal explore the effects on viral infection of introducing short dsRNAs of various sequences and forms. They focus initially on a blunt-ended ds mimic of naturally occurring miR29b-1(*), which inhibited three unrelated viruses. This inhibition was dependent on the cellular RIG-I PRR and interferon production, and was only observed with the RNA as a blunt-ended duplex. The evidence then assembled to elucidate the particular sequence requirements for this effect is somewhat contradictory. On the one hand, the effect is clearly sequence dependent as a randomized version of the mimic had no effect, but equally, several other unrelated sequences also had the same effect. However this latter effect was independent of the RIG-I interferon pathway. So the title of the m/s feels a little misleading in that (a) by no means all short ds blunt RNAs have the inhibitory effect (the title implies they all do) and (b) those that do have an inhibitory effect don’t all do so via the RIG-I interferon system. That a molecule which induces IFN should be antiviral is not really surprising, though all the viruses tested (Ad5, influenza A, SARS-CoV-2) do express functions that mitigate the effects of IFN. Perhaps more interesting then is the later work in the m/s which revealed other short dsRNAs that are inhibitory to Ad5 without needing the IFN system. It would be a valuable addition to show whether or not these dsRNAs can - like the miR29b-1 mimic - also inhibit a wide range of viruses, or whether they are now acting in a sequence-restricted manner.

The manuscript is easy to read, and the experiments reported all appear to have been done with the appropriate rigor and reproduction. The problematic issue is that whilst the m/s describes an interesting phenomenon, and a considerable amount of work has been done then to explore and understand that phenomenon, the end point is rather unsatisfactory as no such understanding has emerged.

Line 726: I think the intended meaning requires ‘postulate’ rather than ‘stipulate’

Author Response

Rev 1:

In their m/s, Kannan etal explore the effects on viral infection of introducing short dsRNAs of various sequences and forms. They focus initially on a blunt-ended ds mimic of naturally occurring miR29b-1(*), which inhibited three unrelated viruses. This inhibition was dependent on the cellular RIG-I PRR and interferon production, and was only observed with the RNA as a blunt-ended duplex. The evidence then assembled to elucidate the particular sequence requirements for this effect is somewhat contradictory. On the one hand, the effect is clearly sequence dependent as a randomized version of the mimic had no effect, but equally, several other unrelated sequences also had the same effect. However this latter effect was independent of the RIG-I interferon pathway.

So the title of the m/s feels a little misleading in that (a) by no means all short ds blunt RNAs have the inhibitory effect (the title implies they all do) and (b) those that do have an inhibitory effect don’t all do so via the RIG-I interferon system. That a molecule which induces IFN should be antiviral is not really surprising, though all the viruses tested (Ad5, influenza A, SARS-CoV-2) do express functions that mitigate the effects of IFN. Perhaps more interesting then is the later work in the m/s which revealed other short dsRNAs that are inhibitory to Ad5 without needing the IFN system. It would be a valuable addition to show whether or not these dsRNAs can - like the miR29b-1 mimic - also inhibit a wide range of viruses, or whether they are now acting in a sequence-restricted manner.

AU: We have modified the title of the manuscript, the new title being: “Sequence-specific features of short double-strand, blunt-end RNAs have RIG-I and type 1 interferon-dependent or -independent antiviral effects”.

It is an interesting question to ask via which mechanism(s) the scr20, scr24, scr32 and scr33 mimics exert their anti-viral effects and how broad their anti-viral effects would be, since these mimics, in contrast to miR29b-1* mimic, retained their anti-viral effects against AdV-C5 in A549 RIG-I or IFNAR2 KO cells. Our preliminary results indicate that these scr mimics vary somewhat in their anti-viral potential against viruses other than AdV-C5. Finding out the mechanism(s) which allow these mimics to exert their effects is a project of its own. Therefore, we prefer not to include further experiments with these scr mimics into the current manuscript.

The manuscript is easy to read, and the experiments reported all appear to have been done with the appropriate rigor and reproduction. The problematic issue is that whilst the m/s describes an interesting phenomenon, and a considerable amount of work has been done then to explore and understand that phenomenon, the end point is rather unsatisfactory as no such understanding has emerged.

AU: This manuscript is a basis for further work, an initial description of the complexity underlying anti-viral effects of short, blunt-end dsRNAs. Untangling the precise sequence features giving rise to anti-viral effects of miR29b-1* mimic, and the cellular molecular players involved is large project on its own.

Since the only obvious sequence feature in miR29b-1* has been the central AUAU-tract we had address the importance of the AUAU tract. It alone turned out not to be decisive for the anti-viral effects of this mimic. Determining sequence features of miR29b-1* that track with its RIG-I- and IFNAR2-dependent anti-viral effects requires screening of a large library of scrambled mimics enough of which would give phenotypes comparable to that of miR29b-1* for comparative assessment. Although our initial library of 33 scrambled mimics was too small to achieve this goal, it nonetheless revealed that not all short, blunt-end dsRNAs exert their anti-viral effects via the same molecular mechanism.

Line 726: I think the intended meaning requires ‘postulate’ rather than ‘stipulate’

AU: yes, indeed “postulate” is the right word here, thank you for pointing this out.

Reviewer 2 Report

The manuscript by Kannan et al explores role of blunt-end dsRNAs as antiviral agents. This is a well written manuscript in which authors identified RIG-1 or INF pathway as a responsible for antiviral effect of miR29b-1 mimic. The data authors provided are well thought of and convincing, however there are some issue that should be resolved prior publication. Please see the comments.

Major comments

1.    The authors propose the following title “Short Double-Strand, Blunt-End RNAs Inhibit Adenovirus, Influenza A Virus and SARS-CoV-2 Infections Depending on RIG-I and Type 1 Interferon”, however they provide concise data only for AdV-C5. There are no data regarding SARS-CoV-2 or IAV infection in the context of RIG-1 or INF. Therefor I strongly suggest to remove IAV and SARS-CoV-2 from the title, because stated as it is can be considered misleading. Otherwise, specific data regarding RIG-1 and INF in the context of SARS-CoV-2 and IAV infection should be provided.

2.    By performing experiment with randomised short blunt-end dsRNAs the authors provide evidence that antiviral effect observed with these molecules can occur directly or indirectly through RIG-I and IFN, or is independent of RIG-I and IFNAR2. Thus I would suggest to reconsider the title and precise that miR29b-1* inhibit adenovirus infection depending on RIG-1 and interferon, and not all blunt-end dsRNAs since this is not the case, as could be understood from the current title.

3.    Please rephrase the sentence in lines 72-76. It is a bit difficult to follow as it is now written.

4.    Figure 1.B Placed as it is now it is difficult to understand the legend corresponding to the graph. Please make it more clear.

5.    Figure 1.D Please clarify designation of black squares.

6.    Can authors please elaborate in more details why they used pVI and not some of the early adenovirus genes to address infection efficiency?

7.    Authors themselves identified potential cytotoxicity of miR29b-1*. Can they comment how this influences interpretation of the virus infectivity data.

8.    The authors present very convincing data on antiviral effect of miR29b-1* against SARS-CoV-2 and IAV viruses. Effect observed on these two viruses in much more pronounced than on AdV-C5. Main difference between SARS-CoV-2, IAV and AdV-C5 is that the first two are RNA viruses, while the later one is DNA virus. Is it possible that the difference in antiviral effect of miR29b-1*could lay in different infection mechanism of studied viruses? Can authors comment on that?

Minor comments

1.    Under the methods section the authors stated that they used approximately 1.5 infectious unit of AdV-C5 for infection. It is quite unusual not to give an exact number of viral particles used for the experiment. Next they left virus for 7h on cells before removing it. Can they please explain what was the reasoning for this experimental set up, having in mind that adenovirus enters the cells within 1h?

Author Response

Rev 2:

The manuscript by Kannan et al explores role of blunt-end dsRNAs as antiviral agents. This is a well written manuscript in which authors identified RIG-1 or INF pathway as a responsible for antiviral effect of miR29b-1 mimic. The data authors provided are well thought of and convincing, however there are some issue that should be resolved prior publication. Please see the comments.

Major comments

1.    The authors propose the following title “Short Double-Strand, Blunt-End RNAs Inhibit Adenovirus, Influenza A Virus and SARS-CoV-2 Infections Depending on RIG-I and Type 1 Interferon”, however they provide concise data only for AdV-C5. There are no data regarding SARS-CoV-2 or IAV infection in the context of RIG-1 or INF. Therefor I strongly suggest to remove IAV and SARS-CoV-2 from the title, because stated as it is can be considered misleading. Otherwise, specific data regarding RIG-1 and INF in the context of SARS-CoV-2 and IAV infection should be provided.

AU: We have modified the title of the manuscript, the new title being: “Sequence-specific features of short double-strand, blunt-end RNAs have RIG-I and type 1 interferon-dependent or -independent antiviral effects”.

2.    By performing experiment with randomised short blunt-end dsRNAs the authors provide evidence that antiviral effect observed with these molecules can occur directly or indirectly through RIG-I and IFN, or is independent of RIG-I and IFNAR2. Thus I would suggest to reconsider the title and precise that miR29b-1* inhibit adenovirus infection depending on RIG-1 and interferon, and not all blunt-end dsRNAs since this is not the case, as could be understood from the current title.

AU: We have modified the title of the manuscript (see answer to the major comment 1)

3.    Please rephrase the sentence in lines 72-76. It is a bit difficult to follow as it is now written. 

AU: We have rephrased and split the sentence in lines 72-76 to two different sentences. The modified text is:

Pioneering as well as large scale siRNA screens have been conducted with several different viruses, for example the negative-stranded segmented RNA virus influenza A virus [6-13], the positive-sense RNA viruses severe acute respiratory syndrome coronavirus (SARS-CoV) and SARS-CoV-2 [14-16], and the double-stranded DNA virus adenovirus (AdV) [2,10,17]. However, there has generally been a limited overlap in the pro- or anti-viral genes identified for a given virus in the different screens [18-20].

4.    Figure 1.B Placed as it is now it is difficult to understand the legend corresponding to the graph. Please make it more clear.

AU: We have modified the legend to Fig. 1B. It now reads as following:

A549 cells transfected with miR29b-1* mimics from different vendors display different infection phenotypes. The blunt-end dsRNA miR29b-1* mimics from Qiagen (Qiagen 29b-1*) and Microsynth (Micros. blunt 29b-1*) reduced AdV-C5 infection efficiency, but Dharmacon (Dharm. 29b-1*) or Microsynth (Micros. overh 29b-1*) dsRNA mimics with 3’ overhangs did not significantly deviate from the results obtained with the Rand. Shown are mean values from three technical replicates. The error bars represent standard deviations. The left and right panels are from two independent experiments.

5.    Figure 1.D Please clarify designation of black squares.

AU: We have added the following sentence to the legend of Fig.1D:

The titer of medium-associated progeny from miR29b-1* mimic-transfected cells is highlighted as black squares.

6.    Can authors please elaborate in more details why they used pVI and not some of the early adenovirus genes to address infection efficiency?

AU: Although both early and late protein expressions are valid indicators for infection efficiency, we use late protein expression for infection efficiency assays here, since progression to late protein expression covers more steps of the infection cycle than early gene expression. The former measures effects on entry, early gene transcription and protein translation, virus DNA replication, late promoter activity and translation of late viral mRNAs, whereas the latter scores interference with virus entry, specific early promoter activity and mRNA translation. Of note, we have also analyzed effect of the miR29b-1* mimic transfection on the kinetics of early protein E1A expression, and in the mimic-transfected cells the E1A expression lagged behind that seen in control cells (data not shown).

7.    Authors themselves identified potential cytotoxicity of miR29b-1*. Can they comment how this influences interpretation of the virus infectivity data. 

AU: Many ISGs promote apoptosis. In addition, activated constitutive IRF3 (without its transcriptional function) can trigger RIPA (RLR-induced IRF3-mediated pathway of apoptosis) killing infected cells by apoptosis. Thus, activation of IRF3 and induction of pro-apoptotic ISGs is one explanation for the reduced cell numbers in miR29b-1*-transfected cells, and this of course could impact virus progeny production.

However, it may not be the only explanation. Gene expression profiling of miR29b-1*-transfected cells indicated the downregulation of several cell cycle-associated genes (Fig. 3A). Hence, the ~ 50% reduction in cell numbers upon miR29b-1* transfection could be due to slowing down of the cell cycle. Although the reduced cell numbers in miR29b-1*-transfected cells was commonly seen (although not to the same extent in every experiment), reduction was not exceedingly drastic at the time points used for analyses.

Currently we do not know the molecular mechanism(s) how the miR29b-1* transfection-induced anti-viral state reduces AdV-C5 infection. Of note, the reduction of cell numbers in miR29b-1* mimic-transfected human diploid fibroblasts was rather small (Fig. 1C) and miR29b-1* mimic had strong anti-viral effects against IAV without drastically reducing the cell numbers (Fig. 2). This implies that cell toxicity effects are unlikely the only reason for anti-viral activity of the miR29b-1* mimic.

8.    The authors present very convincing data on antiviral effect of miR29b-1* against SARS-CoV-2 and IAV viruses. Effect observed on these two viruses in much more pronounced than on AdV-C5. Main difference between SARS-CoV-2, IAV and AdV-C5 is that the first two are RNA viruses, while the later one is DNA virus. Is it possible that the difference in antiviral effect of miR29b-1*could lay in different infection mechanism of studied viruses? Can authors comment on that? 

AU: In our current study we have only tested three viruses. So at this point we cannot say whether the miR29b-1* mimic in general is more effective against RNA viruses than DNA viruses, especially since at this point we do not know the exact molecular mechanisms how miR29b-1*-transfection counteracts AdV-C5, IAV and Sars-CoV-2 infections and what are the relevant host factors behind the anti-viral effects against these viruses.

Minor comments

1.    Under the methods section the authors stated that they used approximately 1.5 infectious unit of AdV-C5 for infection. It is quite unusual not to give an exact number of viral particles used for the experiment. Next they left virus for 7h on cells before removing it. Can they please explain what was the reasoning for this experimental set up, having in mind that adenovirus enters the cells within 1h? 

AU: As stated in the Material and Methods section “Transfections and infections”, the standard infection protocol for AdV-C5 infection assays was inoculation of cells with two-three different two-fold dilutions of input virus so that infection efficiencies were on linear range with respect to the input virus at 24 h – 26 h post virus addition. These standard assays were continuous infections, without removal of the input virus, and several different virus preparations were used in these assays. The moi of approximately 1.5 infectious units and 7 h incubation with input virus refers only to the experiment shown in Fig. 1D, where the AdV-C5 progeny production from control vs mimic-transfected cells was measured. We used the word “approximately” here because we did not determine the exact cell numbers in the different samples at 24 h post mimic-transfection, ie at the time when virus was added to the cells. The moi used was calculated assuming the same cell numbers in all the transfections.

In the experiment shown in Fig.1D, the inoculum was left on cells for 7 h. Although the bulk of AdV-C5 arrives at the nuclear envelope about 1 h after binding to the cell, not all viruses in the inoculum bind to cells within 1 h. In fact, by using radioactively labeled viruses, we had described earlier that only about 2% of input viruses bind to cells within an hour of virus addition (Meier et al., JCB 2002, PMID 12221069). Successful virus entry into cells continues for an extended time, until about 15 h post virus addition, although this time point depends on the initial moi. In Fig. 1D, we used 7 h for virus inoculation, in order to reduce the input virus amount. We furthermore carefully removed then the input virus to minimize interference of the inoculum with titration of progeny particles.

Reviewer 3 Report

The manuscript ‘Short Double-Strand, Blunt-End RNAs Inhibit Adenovirus, Influenza A Virus and SARS-CoV-2 Infections Depending on RIG-I and Type 1 Interferon’ by Kannan and coworkers describes the serendipitous identification and characterisation of cellular miRs mimics that can induce an antiviral state that inhibits the replication of various viruses.  This effect is induced only by blunt-ended double-stranded miR mimics and is dependent on expression of INFARs and RIG-I. This is a very interesting and new finding that can shed light on inducing an antiviral state in cells that can inhibit replication of both DNA and RNA viruses.

The manuscript is well written and does covers the existing literature in this field. The experiments are clearly described and the  data shown are definitely sufficient to support the conclusions of the authors.

There are only a few suggestions for improvements.

11 )      It is not clear how the short double-strand, blunt-end RNAs affect cell viability: Do the cells recover from the induction of this antiviral state?

22 )      I would suggest to replace the phrase ‘primary fibroblasts’ by either ‘diploid fibroblasts’ or ‘non-transformed fibroblasts’ as these names describe the cells much better. 

Author Response

Rev 3:

The manuscript ‘Short Double-Strand, Blunt-End RNAs Inhibit Adenovirus, Influenza A Virus and SARS-CoV-2 Infections Depending on RIG-I and Type 1 Interferon’ by Kannan and coworkers describes the serendipitous identification and characterisation of cellular miRs mimics that can induce an antiviral state that inhibits the replication of various viruses.  This effect is induced only by blunt-ended double-stranded miR mimics and is dependent on expression of INFARs and RIG-I. This is a very interesting and new finding that can shed light on inducing an antiviral state in cells that can inhibit replication of both DNA and RNA viruses. The manuscript is well written and does covers the existing literature in this field. The experiments are clearly described and the  data shown are definitely sufficient to support the conclusions of the authors.

AU: we thank the reviewer for this assessment.

There are only a few suggestions for improvements.

11 )      It is not clear how the short double-strand, blunt-end RNAs affect cell viability: Do the cells recover from the induction of this antiviral state?

AU: At this point, we do not know exactly why the cell numbers are reduced in miR29b-1* mimic-transfected A549 cells. This reduction varies between cells, ie. the reduction was rather minimal in the non-transformed human fibroblasts (Fig. 1C), although the experiments in these cells required longer incubations. The reduction in cell numbers in miR29b-1* mimic-transfected A549 cells tends to get more drastic with increasing time, so these cells do not very well recover from the effects of this mimic. However, we have not tested the recovery after extended incubations and we cannot exclude that some cells do in fact recover. As mentioned in the response to reviewer 2, at this point the available data do not indicate that cell toxicity is a major reason for anti-viral effects of the miR29b-1* mimic.

22 )      I would suggest to replace the phrase ‘primary fibroblasts’ by either ‘diploid fibroblasts’ or ‘non-transformed fibroblasts’ as these names describe the cells much better. 

AU: We have changed the phrase «primary fibroblasts» to «non-transformed fibroblasts».

Round 2

Reviewer 2 Report

All comments were properly addressed by the author's notes and the manuscript was updated accordingly. The revised manuscript is in good form and will be interesting reading for other researchers in the field. Thus I recommend it's publication.